# The Synergistic Mechanism of Photosynthesis and Antioxidant Metabolism between the Green and White Tissues of *Ananas comosus* var. *bracteatus* Chimeric Leaves

**DOI:** 10.3390/ijms24119238

**Published:** 2023-05-25

**Authors:** Dongpu Lin, Xuzixin Zhou, Huan Zhao, Xiaoguang Tao, Sanmiao Yu, Xiaopeng Zhang, Yaoqiang Zang, Lingli Peng, Li Yang, Shuyue Deng, Xiyan Li, Xinjing Mao, Aiping Luan, Junhu He, Jun Ma

**Affiliations:** 1College of Landscape Architecture, Sichuan Agricultural University, Chengdu 611100, Chinalynndtdt@163.com (L.P.);; 2Tropical Crop Genetic Resources Institute, Chinese Academy of Agricultural Science, Haikou 571101, China

**Keywords:** *Ananas comosus* var. *bracteatus*, chimeric leaves, photosynthesis, antioxidant system, synergistic mechanism

## Abstract

*Ananas comosus* var. *bracteatus* (*Ac. bracteatus*) is a typical leaf-chimeric ornamental plant. The chimeric leaves are composed of central green photosynthetic tissue (GT) and marginal albino tissue (AT). The mosaic existence of GT and AT makes the chimeric leaves an ideal material for the study of the synergistic mechanism of photosynthesis and antioxidant metabolism. The daily changes in net photosynthetic rate (NPR) and stomatal conductance (SCT) of the leaves indicated the typical crassulacean acid metabolism (CAM) characteristic of *Ac. bracteatus*. Both the GT and AT of chimeric leaves fixed CO_2_ during the night and released CO_2_ from malic acid for photosynthesis during the daytime. The malic acid content and NADPH-ME activity of the AT during the night was significantly higher than that of GT, which suggests that the AT may work as a CO_2_ pool to store CO_2_ during the night and supply CO_2_ for photosynthesis in the GT during the daytime. Furthermore, the soluble sugar content (SSC) in the AT was significantly lower than that of GT, while the starch content (SC) of the AT was apparently higher than that of GT, indicating that AT was inefficient in photosynthesis but may function as a photosynthate sink to help the GT maintain high photosynthesis activity. Additionally, the AT maintained peroxide balance by enhancing the non-enzymatic antioxidant system and antioxidant enzyme system to avoid antioxidant damage. The enzyme activities of reductive ascorbic acid (AsA) and the glutathione (GSH) cycle (except DHAR) and superoxide dismutase (SOD), catalase (CAT), and peroxidase (POD) were enhanced, apparently to make the AT grow normally. This study indicates that, although the AT of the chimeric leaves was inefficient at photosynthesis because of the lack of chlorophyll, it can cooperate with the GT by working as a CO_2_ supplier and photosynthate store to enhance the photosynthetic ability of GT to help chimeric plants grow well. Additionally, the AT can avoid peroxide damage caused by the lack of chlorophyll by enhancing the activity of the antioxidant system. The AT plays an active role in the normal growth of the chimeric leaves.

## 1. Introduction

*Ananas comosus* var. *bracteatus* (*Ac. bracteatus*) is a tropical monocot belonging to the Bromeliaceae family, which has high ornamental value due to its green–white chimeric leaves and red fruits, and is used as an outdoor decoration and a source of cut flowers [1,2]. The chimeric leaves of *Ac. bracteatus* are composed of central green tissue (GT) and albino tissue (AT). The inhibition of chlorophyll and carotenoid biosynthesis and chloroplast development results in the white color of the marginal tissue of the chimeric leaves [3]. It is regarded that a lack of chlorophyll and carotenoids in plants has a negative impact on photosynthetic functions [4,5]. However, the lack of chlorophyll in the marginal tissue of the chimeric leaves does not destroy the normal growth of the chimeric leaves. As they are completely integrated, not separated individuals, it is uncertain whether the green and white tissues of chimeric leaves have cooperative functions in photosynthesis, and why both the green and white tissues of the chimeric leaves can grow normally is unknown. Understanding of the cooperative mechanism between the chimeric tissues is very important and meaningful to further reveal the growth and development mechanism of plants.

As one of raw materials in photosynthesis, CO_2_ is absorbed and fixed in three main approaches, according to their features of fixation: C_3_, C_4_ and Crassulacean acid metabolism (CAM) [6]. The characteristic of CAM is that stomata open to fix CO_2_ in nocturnal periods, and then CO_2_ is catalyzed by phosphoenolpyruvate carboxylase (PEPC) to form oxaloacetic acid (OAA) with HCO_3_^−^, and then is finally transformed into malic acid stored in the vacuole; during the daytime, malic acid is converted to CO_2_ by NADP-ME and transferred into chloroplast for photosynthetic reactions, and accumulates in the form of starch and soluble sugar [7,8]. Based on these features, Winter and Holtum [9] stated that the most efficient method to determine CAM features in plants is, compared with the traditional observation of stomata in cross sections, investigating net photosynthetic rate (NPR) and stomatal conductance (SCT) using a photosynthesis-measuring device (LI-series), which could provide real and processive indices of alive samples. No matter which approach of CO_2_ utilization is used in plants, the chloroplast is indispensable in the process of photosynthesis. Indices of chlorophyll fluorescence, including minimum fluorescence in light condition (Fo′), maximum fluorescence in light condition (Fm′), photochemical quenching coefficient (qP), non-photochemical quenching coefficient (qN) and relative electron transfer rate of PS II (ETR) can describe the photochemical reaction efficiency of the tissues [10,11]. On the other hand, activities of photosynthetic enzymes, such as PEPC, NADP-MDH, NADP-ME and PPDK, can also provide perspectives on the differences of CO_2_ fixation in plant tissues [12].

O_2_ is also necessary for photosynthesis and respiration in plants, and reacts mainly in chloroplasts and mitochondria where it is spread with active electron transition due to biochemical reactions. This means that reactive oxygen species (ROS), including H_2_O_2_ and O_2_^•−^, are easily produced [13,14]. The chloroplast is one of the main organelles of ROS formation in plants [15], and the albino leaf cells often display ROS accumulation [16,17]. Additionally, ROS can also be derived from stress caused by drought, water logging, salt, high or low temperatures and other environmental factors [18,19,20]. No matter how they are produced, excessive accumulation of ROS has a negative and severe damaging impact on the cell membrane, proteins and other molecular compounds [21]. To avoid the harm caused by oxidants, enzymatic and non-enzymatic systems in plants keep the contents of ROS within a normal range [22]. In the enzymatic system, superoxide dismutase (SOD), peroxidase (POD), catalase (CAT) and enzymes in the AsA-GSH cycle (ascorbate peroxidase, APX; monodehydroasorbate reductase, MDHAR; dehydroascorbate reductase, DHAR; glutathione reductase, GR) play prominent roles in scavenging ROS; whereas in the non-enzymatic system, carotenoids, ascorbic acids, glutathione and other antioxidants alleviate the harm caused by ROS [23,24]. In normal conditions, both the photosynthetic and albino tissues of chimeric leaves show a good state in daily metabolism; it is unknown how the different tissues cooperate in ROS scavenging.

In this study, the cooperative mechanism of photosynthesis and antioxidant systems between the green photosynthetic and white albino tissues of the chimeric leaves of *Ac. bracteatus* were analyzed. These research results could provide insight into the synergistic mechanism of physiological functions between chimeric tissues, which can help further the understanding of the growth and development regulation mechanism in plants.

## 2. Results

### 2.1. Photosynthetic Characteristics of Chimeric Leaves 

In order to reveal the photosynthetic characteristics of the chimeric leaves of *Ac. bracteatus*, the photosynthetic indices of two kinds of chimeric leaves (GrC and WhC, Figure 1) were detected. The values of NPR and SCT of GrCs and WhCs were negative from 12:00 to 20:00, while they turned positive from 21:00 to 8:00. From 9:00 to 11:00, the SCT decreased rapidly and kept at a low level until 21:00 (Figure 2A,B). From 21:00, the SCT of two samples increased significantly and maintained a high level until 9:00 the next morning (Figure 2A). These results confirmed that the photosynthesis of both GrCs and WhCs showed a CAM-like character, i.e., fixing CO_2_ during the night when the stoma was opened and using the stored CO_2_ for photosynthesis during the daytime with closed stoma [25]. The peak value of NPR for the GrCs was about twice that of the WhCs, which indicated that WhCs are inefficient at photosynthesis compared to GrCs (Figure 2A). The SCT of the WhCs was similar to that of the GrCs, and was about two times that of the GrCs at 22:00 (Figure 2B). It was indicated that the low NPR of the WhCs was not caused by a lack of CO_2_, and that the WhCs may have a greater CO_2_ absorption ability during the night. More than 75% of the area of the WhCs comprised white tissue lacking inchlorophyll, which always causes a decrease in photosynthesis in plants [26,27].

In order to further characterize the photosynthesis ability of the GrCs and WhCs, chlorophyll fluorescence parameters were analyzed (Figure 2C). Chlorophyll fluorescence is one of the most popular methods by which to gain information on the state of photosystem II; it has played a major role in understanding the fundamental mechanisms of photosysthesis, the response of plants to environment stress, genetic variation, etc. [28]. There was no significant difference in Fv′/Fm′ and qP between GrCs and WhCs, but the Fv/Fm, ΦPS II and ETR of WhCs were significantly lower than those of GrCs (Figure 2C). These results indicated that the photosynthesis inhibition in WhCs may be directly related to the decrease in PSII photochemical reaction ability and electron transfer rate [29,30]. However, the qN value of the WhCs was significantly higher than that of GrCs, which indicated that the WhCs can dissipate the excess light energy as a form of thermal energy to protect the photosynthetic reaction center from being destroyed by excess light energy [31].

### 2.2. Character of Photosynthetic Pigments Accumulation in Chimeric Leaves

The main differences between the green and white tissue of the chimeric leaves were the photosynthetic pigments contents, which can cause many physiological changes. The central green tissue (GT) and marginal white tissue (AT) of the chimeric leaves were used in the current study. Both the total chlorophyll content and carotenoid content in the GT were about 16 times to that of the AT. It was suggested that not only chlorophyll, but also carotenoid biosynthesis and accumulation, were inhibited in AT. Chlorophyll and carotenoid are important photosynthetic pigments in plants which play important roles in the light harvesting and conversion of photosynthesis. A lack of chlorophyll and carotenoid will inhibit photosynthesis and cause peroxidation damage [32]. The Chla/b ratio of the GT was about 3, while that of the AT was about 0.6 (Figure 3A), which indicated that the biosynthesis of Chla in the AT was more inhibited than Chlb, and that AT is more inclined to absorb short-wave light during photosynthesis. The extremely great reduction in photosynthetic pigments in the AT may directly affect the photosynthetic ability of the leaves.

### 2.3. CO_2_ Utilization Mechanism of the Chimeric Leaves

Because of the CAM character of *Ac* var. *bracteatus*, it was important to reveal the CO_2_ utilization mechanism of the chimeric leaves and the cooperative relationship between the central green and marginal white part of the chimeric leaves. In CAM plants, CO_2_ is fixed by PEPC and stored in malic acid; the reaction is catalyzed by NADP-MDH, NADP-ME and PPDK. The PEPC activity of GT and AT were decreased significantly during the daytime (from about 9:00 to 18:00) and increased at night (from 21:00 to 6:00), and reached the peak value at 3:00 to 6:00 in the early morning. These results further confirmed that *Ac.* var. *bracteatus* is a CAM plant which fixes CO_2_ with PEPC during the night (Figure 3B). NADP-MDH activity of GT and AT showed highly similar changes throughout the whole day (Figure 3C). NADP-MDH activity increased at night, and simultaneously reached maximum at 21:00 while decreasing to minimum at 15:00. However, NADP-ME activity increased during the daytime and decreased before 24:00 (Figure 3D). Additionally, NADP-ME activity of AT kept at a significantly higher level than GT from 6:00 to 15:00. It was suggested that the higher NADP-ME activity in the AT may help release more CO_2_ from malic acid for photosynthetic utilization. According to PPDK activity (Figure 3E), GT and AT showed a similarly changing trend, which increased during the daytime (from 6:00 to 18:00) and decreased during the night (from 18:00 to 6:00). It was suggested that the increase in PPDK activity during the day time helped to promote regeneration of PEP by pyruvate acid, which is a substrate of CO_2_ fixation at night.

In CAM plants, CO_2_ is fixed in malic acid and stored in the vacuole during the night. The malic acid content in the GT and AT increased during the night and decreased during the daytime, which confirms the CAM character of *Ac* var. *bracteatus*. Furthermore, the malic acid content of the AT was significantly higher than in the GT for most of the time (Figure 3F). These results suggest that AT was more effective at CO_2_ fixation and stored more malic acid during the night and kept higher NADPH-ME activity to release CO_2_ from malic acid for photosynthesis during the daytime. However, soluble sugar content (SSC) in the AT was significantly lower than that of GT throughout the whole day (Figure 3G), while the starch content (SC) of the AT was significantly higher than that of the GT (Figure 3H). These results indicated that, although the AT was ineffective at photosynthesis, the photosynthate of the GT was transported to the AT and stored as starch. The export of photosynthate is beneficial to the improvement of photosynthetic efficiency [33,34]. In conclusion, although the AT was ineffective at photosynthesis, it played active roles in keeping the high photosynthesis ability of GT, by acting as a CO_2_ pool and a photosynthate sink.

### 2.4. ROS Equilibrium Mechanism of Chimeric Leaves

It has been reported that the albino tissue is often damaged by the accumulation of ROS [35]. However, although the chimeric leaves of *Ac.* var. *bracteatus* are composed of GT and AT, it can grow and develop normally. Although we have confirmed that the AT can dissipate excess light energy as a form of thermal energy to protect the photosynthetic reaction center, it was still important to reveal the ROS equilibrium mechanism of the chimeric leaves. In order to evaluate the peroxide stress of AT, the contents of O_2_^•−^, H_2_O_2_ and MDA were detected. The contents of O_2_^•−^ and MDA in the AT were significantly lower than those of the GT, but the H_2_O_2_ content in the AT was apparently higher than that of the GT (Figure 4A–C). These results demonstrated that there was no apparent accumulation of ROS in white tissue of the chimeric leaves, and therefore no membrane lipid peroxidation. Regarding antioxidant enzymes, SOD in the AT reached nearly 600 U·g^−1^·min^−1^, about 3 times that of GT (Figure 4D). It was indicated that excessively accumulated O_2_^•−^ was converted into H_2_O_2_ in the AT by high SOD activity. Additionally, activity of CAT and POD in the AT was significantly higher than that of GT, which suggested that AT possessed a relatively stronger antioxidant enzymatic system for scavenging H_2_O_2_. H_2_O_2_ was catalyzed by CAT to produce O_2_ and H_2_O, while POD completed the conversion of H_2_O_2_ into H_2_O [36].

The AsA-GST cycle is important for the scavenging of H_2_O_2_ and avoiding membrane lipid peroxidation, and the APX, MDHAR, DHAR and GR are the most important enzymes in it. AsA acts as an electron donor for APX to remove H_2_O_2_ and produced Monodehydroascorbic acid (MDHA), which can be reduced by monodehydroascorbate reductase (MDHAR) [37]. Using GSH as an electron donor, dehydroascorbate reductase (DHAR) can reduce DHA to AsA, and the GSSG produced can be reduced to GSH by glutathione reductase (GR). Thus, the process of removing H_2_O_2_ and regenerating AsA and GSH is completed [38].

The total ascorbic acid content (TAA) and total glutathione content (TG) were significantly higher in the AT than in the GT, and the ratios of GSH/TG and DHA/TAA were also increased significantly in the AT compared to in the GT (Figure 4F). The activity of APX, MDHAR and GR in the AT was significantly higher than that in the GT, especially the APX activity, which in the AT was about 3 times that in the GT. The activity of DHAR showed no significant difference between the two samples. These results indicated that the AT had a more active AsA-GST cycle to keep the ROS equilibrium (Figure 5).

## 3. Discussion

Studies on CO_2_ fixation and utilization subject to crassulacean acid metabolism (CAM) in plants have been reported [39,40,41]. Generally speaking, the feature of CAM is that CO_2_ is fixed and combined with PEP through catalysis of PEPC enzymes to form OAA, and then OAA is transformed into malic acid and stored in the vacuole, which results in an increase in malic acid content during nocturnal times; conversely, to avoid excessive evaporation of internal moisture, malic acid is catalyzed as CO_2_ to supply photosynthesis without opening stomata during diurnal periods [42,43,44]. Based on the predominant enzymes employed during the daytime decarboxylation of malate leading to intracellular CO_2_ release, CAM plants were divided into two general groups. Among the 1893 species of plants in the Bromeliaceae family, 57% of species exhibit CAM characteristics [45]. In the photosynthesis progress of *Ananas comosus* var. *bracteatus*, the CO_2_ is fixed in malic acid during the night, and the higher NADP-ME activity helps to release CO_2_ from malic acid for photosynthetic utilization during the day. Therefore, we speculate that it is NAD(P)-malic enzyme (ME)-type species, which decarboxylates malate to release CO_2_ and pyruvate, with pyruvate being regenerated to PEP by PPDK [46].

The chimeric leaves of *Ac. bracteatus* are composed of green center tissue and white marginal tissue. It is useful to reveal the relationship between the green photosynthetic tissue and white albino tissue and the cooperative mechanism of the two types of tissues by which they grow normally. The leaf is the main organ for photosynthesis; chlorophylls and carotenoids are located in the chloroplasts of the leaves and can absorb and convert light energy for photosynthesis. Albino leaves might have negative impacts on plant growth and development. The lack of chlorophyll and carotenoids in AT, especially the severe deficiency of Chl a, did lead to a decrease in the photosynthetic capacity of AT. AT had a significantly decreased photochemical reaction rate and electron transport rate, and most of the absorbed light energy was dissipated as thermal power; this might be a protective measure for the photoreaction center of albino leaf tissue [47].

Photosynthetic enzyme activity indicated that AT was not a redundant part, but a pool of photosynthetic carbon compounds. We reasonably inferred that AT plays a role in fixing and storing carbon during the night and supplying CO_2_ in the daytime to cooperate with GT, and helps GT to have a higher photosynthesis rate by improving CO_2_ concentration even though stomatal conductance is low during the daytime. In addition, AT had a higher starch content than that of GT, which indicated that the AT may also function as a storage of photosynthate. The export of photosynthate helped the GT to have a higher photosynthesis rate.

Abnormal albinism always has negative effects on plant individuals, likely leading to a reduction in yield and even death [48]. Absence of photosynthetic pigments always has a domino effect on physiological functions, including stress resistance. For instance, soluble sugar plays an essential role in salt and drought stress, which helps plants avoid further moisture loss by increasing intracellular osmotic pressure, but a low level of soluble sugar probably results in the plant being more vulnerable to outer stress [49]. Additionally, albino tissue, whose chloroplasts and mitochondria produce more ROS because of obstructed electron transfer, are likely weaker and more sensitive to stress than green photosynthetic tissues, even though they keep the ROS level under control with a more active scavenging system [50,51].

However, *Ac. bracteatus* maintained a balance between the formation and elimination of peroxides, keeping MDA, H_2_O_2_ and O_2_^•−^ at a low level to prevent the harm caused by loss of chlorophyll. First of all, AT contained a significantly higher level of AsA and GSH, indicating that AT has an apparently higher antioxidant capacity. The AsA-Gsh cycle is an important way to scavenge peroxides and also the major source of AsA and GSH generation. It has been demonstrated that AT can further improve the ability to scavenge ROS by enhancing the AsA-Gsh cycle. At the same time, AT had considerably higher SOD, CAT and POD activity than GT. We propose that SOD transforms a significant portion of O_2_^•−^ into H_2_O_2_, and a high activity of POD and CAT can scavenge H_2_O_2_.

## 4. Materials and Methods

### 4.1. Plant Materials and Sample Preparation

*Ac. bracteatus* plants were grown in 10 × 10 cm pots and maintained in a semi-structured greenhouse under natural conditions. The healthy current-year chimeric leaves at the fourth layer from the youngest leaves of three-year-old *Ac. bracteatus* in Sichuan Agricultural University research garden were used in this study (Figure 1). Green chimeric leaves (GrC, green tissue ≥ 65%, Figure 1B) and white chimeric leaves (WhC, white tissue ≥ 75%, Figure 1C) were used for measurement of net photosynthetic rate, stomatal conductance and chlorophyll fluorescence parameters every 2 h during a day. The detections were conducted on three leaves from three different plants, and for each leaf the detections were conducted three times at each time point. The proportion of green or white tissue was measured using Photoshop 2020 (Adobe, San Jose, CA, America). Central green photosynthetic tissue (GT) and marginal white albino tissue (AT) were carefully cut from typical chimeric leaves and used for the detection of pigment content, photosynthetic enzyme activity, antioxidant enzyme activity, malic acid, starch content, soluble sugar content, ROS content, antioxidants content and activities of the AsA-GSH cycle. Each determination contained three technical replicates, performed in different leaves mixed from three plants.

### 4.2. Measurement of Photosynthesis and Chlorophyll Fluorescence Parameters 

To examine the differences in photosynthetic ability between GrCs and WhCs, a photosynthesis-measuring device (LI-6400, LI-COR Co., Ltd., Lincoln, Nebraska, America) was used in this assay. GrCs and WhCs were placed in the device with a natural light source and stable CO_2_ flow rate at 500 μmol·s^−1^. Data of NPR (μmolCO_2_·m^−2^·s^−1^) and SCT (molH_2_O·m^−2^·s^−1^) were recorded every 30 min from 9:00 to 8:30 next day. Indices of chlorophyll fluorescence were measured using a Handy FluorCam 1000-H (Photon System Instruments, Brno, Morava, Czech Republic), and calculation followed a previous method [52].

### 4.3. Determination of Photosynthetic Enzyme Activities

The enzyme solution was extracted based on a protocol by Chojak-Koźniewska et al. [53]. Fresh leaf tissues (0.5 g) were homogenized in 6 mL of enzyme-extracting solution (50 mM Tris-HCL, pH 7.5; 10 mM MgCl_2_; 5 mM DTT; 2% (*v*/*v*) PVP; 10% (*v*/*v*) glycerol) and centrifuged at 13,000× *g* at 4 °C for 10 min, and the supernatant was used for measuring enzyme activity.

PEPC activity was examined following the method by Gonzalez et al. [54]. The absorbance of enzyme solution was recorded every 30 s at a wavelength of 340 nm for 3 min after 20 μL phosphoenolpyruvate (PEP, 100 mM) was added.

NADP-ME activity was tested using the protocol of Penninckx et al. [55]. Conversion of MA to pyruvic acid with concurrent NADP^+^ reduction was measured at 340 nm every 30 s for 3 min.

NADP-MDH activity was examined according to Sayre and Kennedy [56]. The mixture contained 100 mM NADH, 1 Mm EDTA, 0.5 Mm oxaloacetic acid (OAA) and distilled water (ddH_2_O). A 3 min record of absorbance at 340 nm was made every 30 s in this period for calculating NADP-MDH activity.

PPDK activity was determined using the protocol of Camp et al. [57]. After components were mixed well, 2 μL 1 mM NADPH was added and an absorbance record was made at 340 nm every 30 s for 3 min.

### 4.4. Determination of Photosynthetic Pigments

A NanoDrop 2000 spectrophotometer (Thermo Scientific, Waltham, MA, USA) was used as the equipment for detecting photosynthetic pigments. The central photosynthetic and marginal albino tissues of the fresh and mature functional chimeric leaves were cut separately for the detection of chlorophyll a (Chl a), chlorophyll b (Chl b) and carotenoid (Car) contents [58]. Fresh leaf tissue (0.2 g) was homogenized with 3 mL acetone and 10 mL ethanol and filtered into a 25 mL brown volumetric flask. Then, 80% acetone was taken as the blank, and the absorbance was measured at the wavelengths of 663 nm, 646 nm and 470 nm.

### 4.5. Assays of Antioxidant Enzyme Activities

Fresh leaf tissue (0.2 g) was homogenized with 1 mL 0.05 M PB (Phosphate Buffer, pH 7.8) and 3 g PVP. The mortar was washed three times with 7 mL PB, transferred into a 10 mL centrifuge tube and centrifuged at 4500 rpm for 20 min at 4 °C. The supernatant was volumed to 10 mL and used for enzyme activity detection.

SOD activity was measured using the methodology of Abassi et al., [59]. Absorbance of the reaction solution was examined using a spectrophotometer at 560 nm every 30 s for 3 min. The SOD activity was expressed as U·g^−1^·min^−1^.

CAT activity was measured following a previous protocol [60]. The absorbance of reaction solution was recorded every 30 s for 3 min at 240 nm, and the CAT activity was expressed as U·g^−1^·min^−1^.

POD activity was examined according to Lee and Kim [61]. The assay mixture contained 10 mM PBS, 20 mM guaiacol, 50 mM H_2_O_2_ and 10 μL enzyme extraction. The reaction began with the addition of H_2_O_2_ and absorbance was recorded every 30 s for 3 min at 470 nm. The enzyme activity is depicted as U·g^−1^·min^−1^.

### 4.6. Assays of Malic Acid, Starch and Soluble Sugar Contents

Content of malic acid was measured using Plant Malic acid (MA) Enzyme-linked immunosorbent assay (ELISA) kit (Yuan Ye, Shanghai, China). Malic acid extract was obtained based on the protocol of Kou et al. [62], and the content was measured at 450 nm using a Multiskan (Thermofisher Co., Ltd., Waltham, MA, USA) according to the manufacturer’s instructions.

Starch content was detected according to the methods of Hansen and Moller [63]. The soluble sugar supernatant was isolated and the content was detected according to the methods of Deng et al. [64].

### 4.7. Determination of H_2_O_2_, O_2_^•−^, MDA, Antioxidants and Enzyme Activities in AsA-GSH Cycle 

Contents of H_2_O_2_**,** O_2_^•−^ and MDA were measured and calculated by following the detailed methods of Hasanuzzaman and Fujita [65]. The contents of ascorbate (AsA), dehydroascorbate (DHA), glutathione (GSH) and glutathione disulphide (GSSG) were measured according to Ozgur et al. [66]. Ascorbate peroxidase (APX), Monodehydroascorbate reductase (MDHAR), dehydroascorbate reductase (DHAR), and glutathione reductase (GR) were extracted and measured based on the method of Pasquariello et al. and Rahman et al. [67,68].

### 4.8. Statistical Analysis

Statistical analyses were performed with SPSS.22, and significant differences were examined at *p* < 0.05 using the *t*-test. Figures were visualized in Graphpad Prism version 8.0.

## 5. Conclusions

This study confirmed that *Ac. bracteatus* is a CAM plant, which fixes CO_2_ by PEP during the night and releases CO_2_ from malic acid for photosynthesis during the daytime. From the perspective of CAM plants, we speculate that *Ac. bracteatus* is an NADP- ME-type species. The AT stored more malic acid than the GT during the night due to higher activities of PEPC, PPDK and NADP-MDH, and released CO_2_ from malic acid during the daytime through higher activity of NADP-ME. The soluble sugar content of the AT was significantly lower than that of the GT, while the starch accumulated more in the AT. It is suggested that AT helps GT to keep a high photosynthetic rate by acting as a CO_2_ pool and photosynthate storage. The apparent reduction in photosynthetic pigments in the AT resulted in a higher accumulation of H_2_O_2_, but the ROS equilibrium in the AT was kept to maintain the normal growth of albino leaves by enhanced activities of SOD, POD, CAT and a more active AsA-GSH cycle.

## Figures and Tables

**Figure 1 ijms-24-09238-f001:**
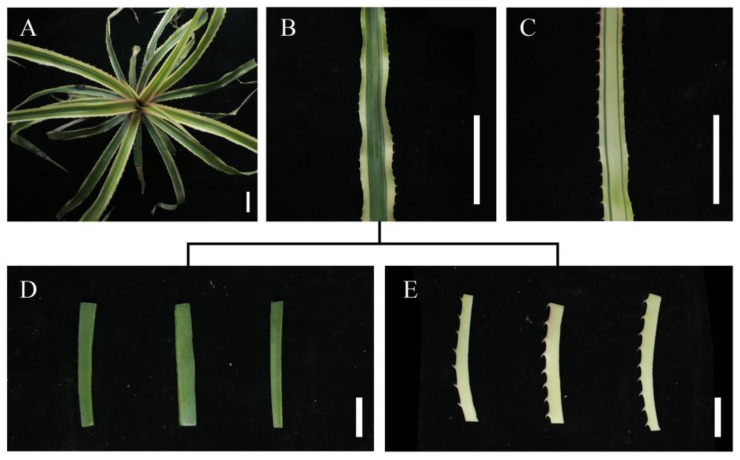
Leaf samples used in the study. Chimeric plant of *Ac. bracteatus* (**A**). Green chimeric leaves (GrC, (**B**)) and white chimeric leaves (WhC, (**C**)). Central green photosynthetic tissue (GT, (**D**)) and marginal white albino tissue (AT, (**E**)) cut from the chimeric leaves. The bar of first row = 5 cm, and the second row = 1 cm.

**Figure 2 ijms-24-09238-f002:**
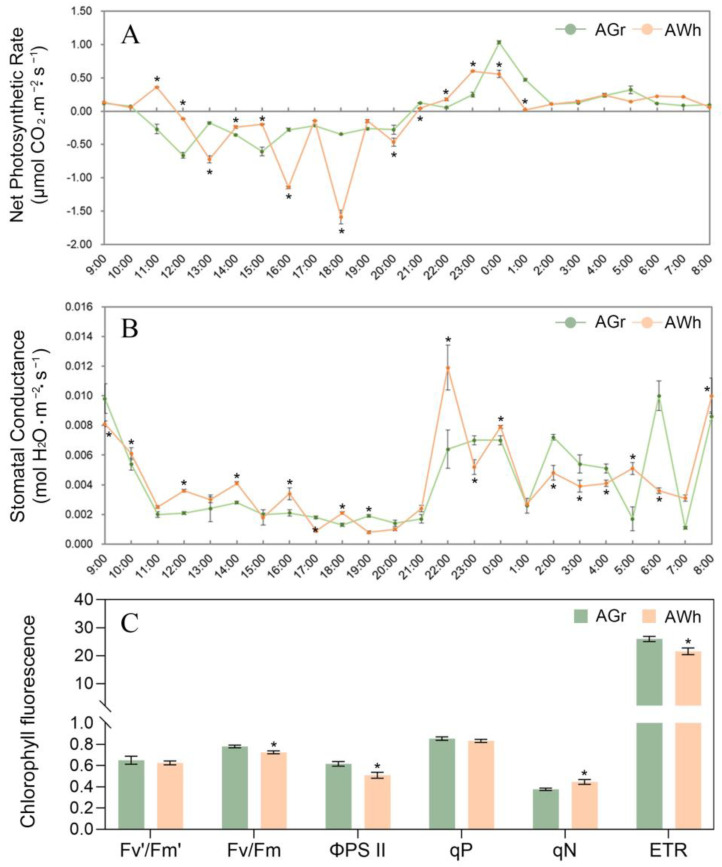
Photosynthetic characters of the green chimeric leaves (GrC) and white chimeric leaves (WhC). Daily net photosynthetic rate of GrC and WhC (**A**). Daily stomatal conductance of GrC and WhC (**B**). Chlorophyll fluorescence parameters of GrC and WhC (**C**). Asterisk in the graphs represents significant difference according to *t*-test (*p* < 0.05).

**Figure 3 ijms-24-09238-f003:**
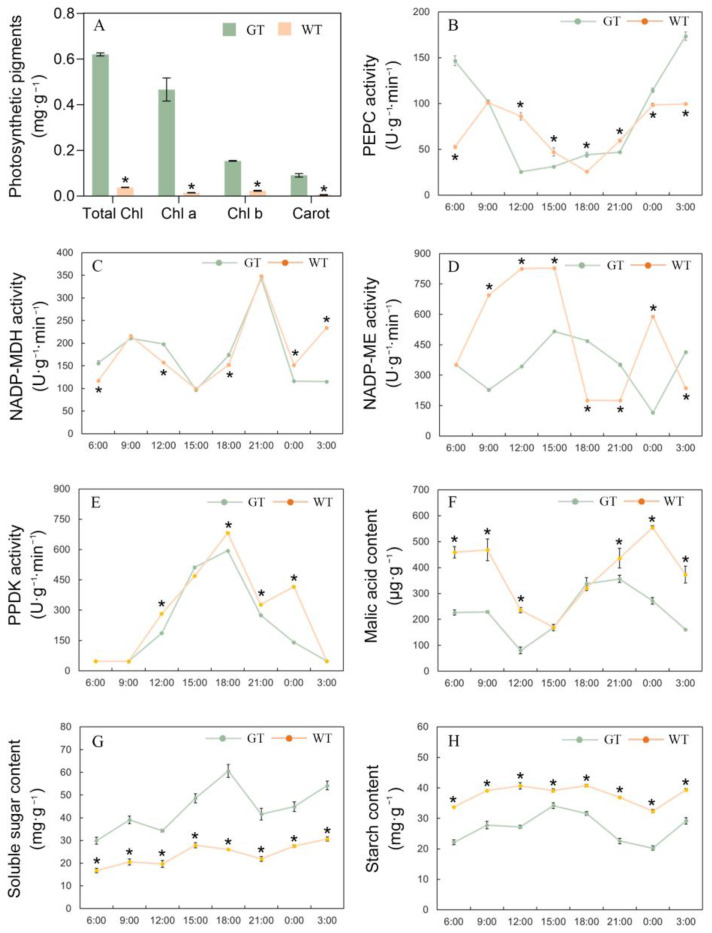
Photosynthetic characters of the central green photosynthetic tissue (GT) and marginal white albino tissue (AT) of the chimeric leaves. Photosynthetic pigment contents in the GT and AT (**A**). Activities of PEPC enzyme in the GT and AT (**B**). Activities of NADP-MDH enzyme in the GT and AT (**C**). Activities of NADP-ME enzyme in the GT and AT (**D**) and Activities of PPDK enzyme in the GT and AT (**E**). Malic acid content in the GT and AT (**F**). Soluble sugar content in the GT and AT (**G**). Starch content in the GT and AT (**H**). Asterisks in the graphs represents significant difference by *T*-test (*p* < 0.05).

**Figure 4 ijms-24-09238-f004:**
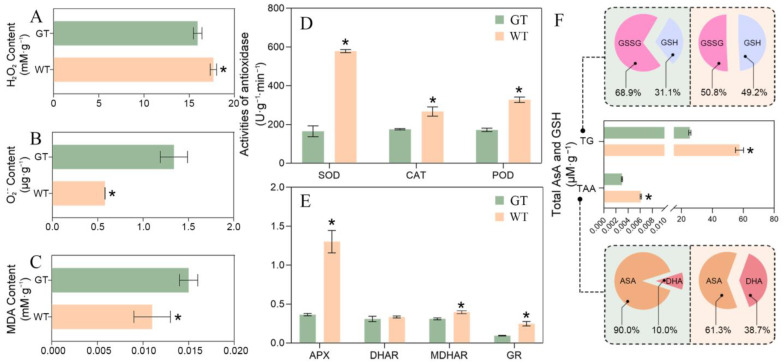
The ROS equilibrium of the central green photosynthetic tissue (GT) and marginal white albino tissue (AT) of the chimeric leaves. H_2_O_2_ content in the GT and AT (**A**). O_2_^•−^ content in the GT and AT (**B**). MDA content in the GT and AT (**C**). Activity of SOD, POD and CAT (**D**). Activity of enzymes in the AsA-GSH cycle (**E**). Contents of antioxidants in the AsA-GSH cycle (**F**). Asterisks in the graphs represents significant difference by *t*-test (*p* < 0.05). TAA, total ascorbic acid content. TG, total glutathione content.

**Figure 5 ijms-24-09238-f005:**
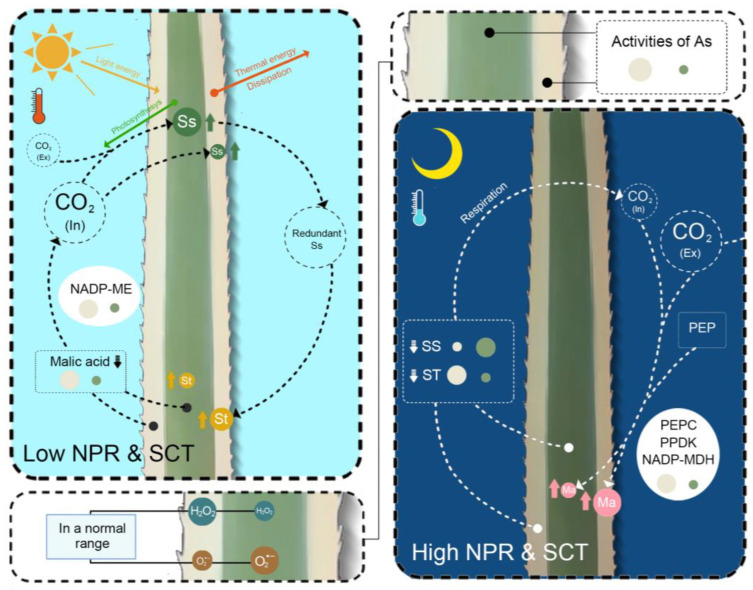
A conceptual model of physiological functions of the central green photosynthetic tissue and marginal white albino tissue of the chimeric leaves of *Ac. bracteatus*. Different sizes of the same colored circle represent a significant difference in values. PEP, phosphoenolpyruvate. Ss, soluble sugar. St, starch. NPR, net photosynthetic rate. SCT, stomatal conductance. As, antioxidant system. In, internal. Ex, External. Ma, malic acid.

## Data Availability

Not applicable.

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
