# Peer review of "The Synergistic Mechanism of Photosynthesis and Antioxidant Metabolism between the Green and White Tissues of Ananas comosus var. bracteatus Chimeric Leaves"

_ijms, 2023, doi:10.3390/ijms24119238_

Round 1

Reviewer 1 Report

Lin et. al attempts to find the function of differently coloured tissues within the leaves of Ananus bracteata utilizing the standard photosynthesis measurements with enzyme activity measurements. The manuscript is well-written and the results are presented coherently. I have the following comments to improve the manuscript:

1.     The use of the abbreviation “WT” might be misleading as it usually denotes wild-type plants. I would suggest changing “WT” to “AT” (Albino Tissue) which is more appropriate to the work

2.     The measurements of photosynthetic parameters were performed in the month of July, according to the method section. But the relevance of this is not mentioned anywhere in the manuscript

3.     Figure 1: How many plants and individual leaves were used for the measurements? Were the measurements performed in the same leaf over time? At what leaf-age the experiments were performed? What do the error bars in the graph represent?

4.     Figure 2: The same questions for the comment above apply here. In addition, for Figure 2C, how many replicates were used?

5.     Figure 3: Questions for Figure 2 and Figure 3 apply here

6.     Line 370: It is written, “kind of CAM plant”. I would suggest speculating on the type of CAM plant and elaborating based on the results, rather than leaving this as a vague statement

Reviewer 2 Report

Article review by Dongpu Lin, Xuzixin Zhou, Huan Zhao, Xiaoguang Tao, Sanmiao Yu, Xiaopeng Zhang, Yaoqiang Zang, Li Yang, Shuyue Deng, Xiyan Li, Xinjing Mao, Aiping Luan, Junhu He and Jun Ma: «The synergistic mechanism of photosynthesis and antioxidant 2 metabolism between the green and white tissues of Ananas 3 comosus var. bracteatus chimeric leaves».

By the authors, using the leafy chimeric plant Ananas comosus var. Bracteatus, the mechanism of interaction of photosynthesis and antioxidant systems between green photosynthetic and white albino leaf tissues has been analyzed. The presence of green photosynthetic tissue (GT) and marginal white albino tissue (WT). makes chimera leaves an ideal material for studying the synergistic mechanism of photosynthesis and antioxidant metabolism. Diurnal changes in the net photosynthesis rate (NPR) and stomatal conductivity (SCT) of leaves indicate a typical metabolism ITSELF in Ac. Bracteatus. The fixation of CO2 during the night period and the release of CO2 from malic acid during the day are shown. The authors believe that WT helps GT maintain high photosynthetic activity by acting as a CO2 pool and photosynthetic storage. In addition, WT maintained the peroxide balance by strengthening the non-enzymatic antioxidant system and the antioxidant-enzymatic system to avoid damage by antioxidants. The activity of ascorbic acid (AsA) and glutathione (GSH) reduction cycle enzymes (except DHAR) and superoxide dismutase (SOD), catalase (CAT), peroxidase (POD) was increased, apparently, for normal leaf growth.

This study showed that although the mass of chimeric leaves was ineffective due to a lack of chlorophyll, but it can interact with GT, working as a CO2 supplier and photosynthesis repository to enhance GT's photosynthetic ability and help chimeric plants grow well.

I think that this work should be accepted for publication in the journal.

Author Response

Thank you for your love of this article. It is an honor to have been inspired by you during the article submission process, which will make me more passionate about scientific research.

Reviewer 3 Report

1) Were the chimeric leaves taken from the same layer or from different ones? It is known that the leaves of different tiers differ in their physiological activity.

2) Have you studied non-chimeric green pineapple leaves to control your research? It is possible that the excised parts of the leaf differ not only in their pigment composition, but also in the location of pathways and the redistribution of metabolites in general. Since you took central green tissue and lateral white tissue for research, it cannot be argued that the above differences are not related to the physiological activity of leaf regions. I think that in this study, some kind of control sample was needed with which to compare.

3) The captions on the drawings are very small, please make them larger.

4) It is not clear from the Materials and methods section, how the assessment of chlorophylls and carotenoids was carried out and on what equipment. Please add a short description.

5) How much time passed from the moment of separation of different leaf tissues to various biochemical analyzes? Could this time affect the results? Have samples been weighed? How and where was the cultivation carried out, was temperature and humidity controlled during cultivation?

6) Why were the green and white chimeric leaves studied, if tissue sampling, as far as I understand, was carried out only for green chimeric leaves?

7) It is necessary to decipher all abbreviations in the captions to figures 4 and 5, for example, Fig. 4: TG, TAA, etc.; Fig. 5: As, Ma.

Round 2

Reviewer 3 Report

Authors improved the manuscript. I do not have any comments.